# Connectivity of Ephemeral and Intermittent Streams in a Subtropical Atlantic Forest Headwater Catchment

**Alondra B. A. Perez** [1,*] , **Camyla Innocente dos Santos** [1] , **João H. M. Sá** [1] , **Pedro F. Arienti** [1] **and Pedro L. B. Chaffe** [2]

[1] Department of Sanitary and Environmental Engineering, Graduate Program of Environmental Engineering, Federal University of Santa Catarina, Florianópolis 88040-970, Brazil; camylainnocente@gmail.com (C.I.d.S.); Jhenriquemsa@hotmail.com (J.H.M.S.); pedro.arienti@posgrad.ufsc.br (P.F.A.)

[2] Department of Sanitary and Environmental Engineering, Federal University of Santa Catarina, Florianópolis 88040-970, Brazil; pedro.chaffe@ufsc.br

[*] Correspondence: alondrabaperez@gmail.com

**Abstract:** Stream network extension and contraction depend on landscape features and the characteristics of precipitation events. Although this dependency is widely recognized, the interaction among overland-flow generation processes, drainage active length, and frequency in temporary streams remains less understood. We studied a forest headwater catchment with wide variation in soil depth to investigate the runoff generation processes that lead to the occurrence of ephemeral and intermittent flow and connectivity between hillslope and outlet. We used low-cost equipment to monitor the variation in the length of the active drainage network and to measure the water table development. The flow in the channels can develop even under light rainfall conditions, while the connectivity is controlled by antecedent wetness, total precipitation, and active contribution area thresholds. Runoff permanence and fragmentation were related to soil depth variation; flow being usually more disconnected due to deeper water tables in deeper soil locations. Our findings emphasized the impact of soil structure on runoff generation in hillslopes and can be useful in the management of the most active areas and their impact on the quality of available water.

**Keywords:** overland flow; intermittent stream; hydrological connectivity; headwater catchment

## 1. Introduction

The estimation of runoff depends on our understanding of the flow paths that water takes to the catchment outlet [1–3]. Among the headwater's drainage network, many channels that control the hillslope hydrological connectivity [2] are ephemeral or intermittent [4]. Intermittent rivers are coupled aquatic-terrestrial systems that flow continuously only at certain times of the year and support a unique biodiversity [5]. While in low-flow periods alternating dry segments can create a discontinuity of the stream flow [6], in wet seasons the active drainage network can expand and become several times larger than the perennial drainage network [7–9]. The quantity and the quality of the streamflow may be influenced by the hydrographic network expansion rate [10] as it influences the transport of water, sediments, and nutrients in small catchments [11] and shapes local biodiversity and ecosystem processes [12]. For example, the refilling of ephemeral channels can affect the path of nutrients and organic matter that accumulate during dry periods, releasing large amounts of dissolved organic carbon [13].

The actual extent of the active hydrography is driven by precipitation events [14] and adjusts dynamically as landscapes become wet or dry [3,15]. Soil characteristics and topography are also important controls in the overland-flow generation processes [16]. The expansion and contraction of

the network length can follow a similar pattern among events, implying localized controls, such as drainage area and slope of the channel [8]. Different portions of the stream channel can change from losing to gaining reaches affecting the extension and connection of the drainage network. Part of those dynamics are controlled by the transmissivity characteristics of the channel [17]. It is possible that a low-intensity rain will generate runoff, but water will be retained or infiltrate in the channel due to local variation in transmissivity. On the other hand, if the rain occurs long enough, the amount of water available will overcome the moisture deficit and the stream will be connected along the slope [18,19]. Those controls over the flow continuity determine the efficiency in the transport of water, sediments, and nutrients [11] and the connection between zones of the catchment [2].

The dynamic of temporary flows depends on the complex interplay of the landscape attributes (e.g., slope, soil depth, lithology, land use) and the characteristics of the precipitation events. While in some environments the temporary flow results from the expansion of the perennial drainage network [3], in other areas channel network expansion and connection may be highly variable. The activation of the flow path in Mediterranean climate mountains, for example [16], was attributed to the interaction between precipitation and the great variation in the saturated hydraulic conductivity ($K_s$) of the soil. Where $K_s$ gradually decreases, deeper water tables are formed. The fast decrease of $K_s$ in the superficial layers of soil may promote the formation of a shallow, suspended, and temporary water table [18]. The same characteristic has been observed in other highly seasonal humid climate basins (e.g., a tropical lowland forest in Panama [3], a humid subtropical pine forest in USA [20], and a grassland area in Australia [18]). In a subtropical, humid, forested hillslope, the beginning of runoff was related to the total rainfall threshold necessary to fill the depressions of the soil-bedrock interface [21,22].

Thus, the dynamics of flowing stream networks is a visible reflection of the underground hydrological processes, which are non-visible and difficult to measure [17,23]. The stormflow hydrographs' shape and water quality is affected by the different sources and flow paths (e.g., deep, shallow, or even temporary streams). Most of our knowledge about runoff generation processes has been based in steep landscape with shallow soils or in low relief landscape with deep soils [24]. Those studies are also concentrated on temperate regions, with a limited number of works in other climates and landscapes [25,26]. Controls over the active stream network generally focus on meteorological patterns and topographic characteristics (e.g., [27]), but few studies have related those processes to the variation in soil depth and its structure.

In this work, we investigated overland-flow generation in ephemeral and intermittent streams, in a coastal environment with steep landscape and contrasting soil depths. The area is covered by subtropical Atlantic Forest. The following questions were addressed: (1) How do the characteristics of rain events, topography, and soil structure control the longitudinal dynamics of the drainage network? (2) Under what conditions does hillslope and outlet connectivity develop? (3) Is it possible to identify the runoff generation processes based on the hydrographic behavior? Rainfall data and low-cost monitoring were used to identify the active and connected network length and the variation in the water table level.

## 2. Materials and Methods

### 2.1. Study Site

The Peri Lagoon is the largest freshwater source for public supply on the island of Santa Catarina, Brazil (Figure 1a–c). The area is an important ecosystem for the preservation and regeneration of the Atlantic Forest [28–30], which is considered a biodiversity hotspot for preservation [31]. According to the Köppen-Geiger criteria, the climatic classification of the region is Cfa (defined as humid subtropical climate), with hot summers and rains distributed throughout the year. The average temperature varies between 15 °C and 27 °C. The average annual rainfall is 1700 mm [32].

The Retiro headwater catchment is a northeast hillslope of the Peri Lagoon watershed with an area of 2.65 ha. The average slope of the study area is 38%. Land cover is mainly Atlantic Forest in a

secondary stage (64% of the total area) with great heterogeneity of tree species [30] and reforestation with pines and grasses at the top of the slope (Figure 1c). The reforested area is used for grazing. The drainage network is formed by ephemeral channels, which are defined as portions of the drainage network that activate in direct response to precipitation events [20,33]. In the middle of the hillslope, there is a section characterized as intermittent (Figure 1c—red dotted line), being active in most of the year [34]. The main channel is shallow and well defined up to the middle of the slope. At the end of the intermittent section, there are a lot of cobbles, and the outlet of the channel is completely covered by boulders. The slope flows into a flat region that extends for 100 m to the bank of the lagoon. The soil is characterized as Alic Podzolic Red-Yellow, with medium and clayey texture, moderately drained with rocky areas and strong undulation [35]. The geology is characterized by plutonic and volcanic igneous rocks of precambrian age (neoproterozoic), mainly granites with numerous diabase dikes.

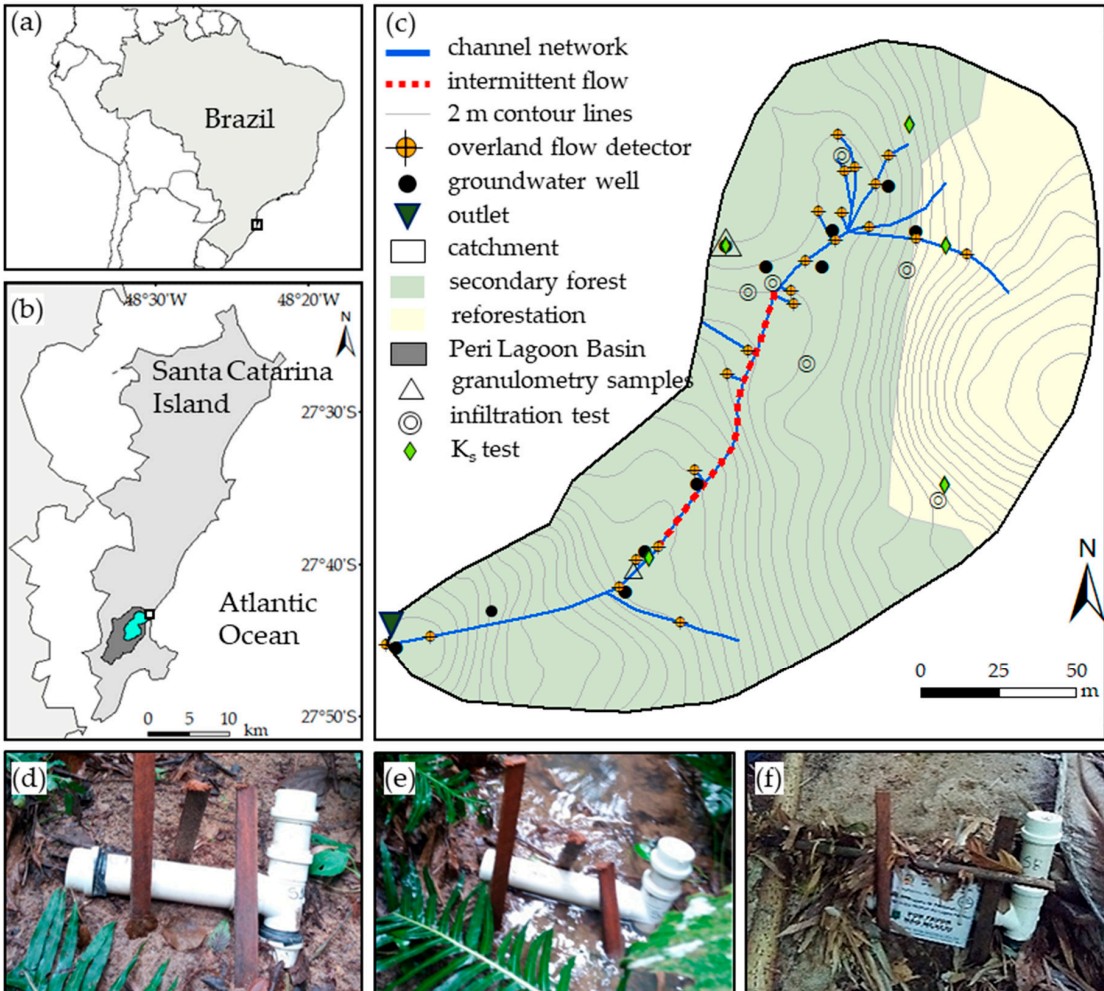

**Figure 1.** The Peri Lagoon watershed is located in Southern Brazil (**a**,**b**). (**c**) Retiro headwater catchment, with the channel system, position of intermittent flow, locations of overland-flow detectors, location of the soil samples, infiltration and saturated hydraulic conductivity (K$_s$) tests, groundwater wells, outlet site, and forest cover. (**d**) View of overland-flow detector installed within the ephemeral channel before the rainfall event, and (**e**) during a rainfall event. (**f**) Litter fall and sediments transported by storm runoff.

Topographic survey was done with a total station to produce a digital elevation model with a resolution of 1 m. The projection system used was Geocentric Reference System for the Americas Datum (*Sistema de Referencia Geocentrico para las Américas*—SIRGAS 2000). The activation frequency of each overland-flow detector was analyzed against the topographic characteristics of the channel and

upstream contribution area. Those areas reflect landscape features, such as topography, geology, soils, climate, and vegetation, which influence the movement of water from its first entry to its final release [1]. Eight topographic features were calculated: Area, channel slope, topographic wetness index (TWI), cumulative area slope, concave curvature, convex curvature, linear curvature, and drainage density.

*2.2. Overland-Flow Detection*

We installed 23 overland-flow detectors (OFDs), the same as developed by Kirkby et al. [36] and described by Vertessy and Elsenbeer [37] and Zimmermann et al. [3]. It was difficult to define where the flow began because the channel heads are diffuse. Thus, the first detector of each channel was placed at the most upstream point where the flow occurrence was visualized. The other detectors were installed where the occurrence of runoff was recorded or where there were traces of its presence, such as litter and sediment drag and deposition (Figure 1f). We could not install the OFDs uniformly spaced in the flow lines and channels due to the heterogeneous channel morphology. For example, due to cobbles and boulders there were only two spots where we could install OFDs in the reach between the wells W8 and W10. We have even seen water flowing beneath those boulders on a few occasions. We installed the OFDs where the channel characteristics allowed and avoided big distances between them. It is not possible to know if the slope was dry before the event with non-automatic data. The ephemeral channels dry quickly after the rainfall event. While OFDs were been installed in the intermittent section, in every campaign we took note on the conditions of the channel reach. There was water flowing in that section most of the time.

The OFDs were installed in contact with the streambed inside each flow line (Figure 1d). When overland flow occurs in the channel, the reservoir is filled (Figure 1e). Based on Zimmermann et al. [3], we made the following assumptions: If one OFD was active and the adjacent downstream one was empty (full-empty situation), we considered that the flow occurred up to the first OFD; if an OFD was empty and the adjacent downstream one was full (empty-full situation) we considered that flow started from the downstream full OFD. We had to assume that if two adjacent OFDs were full (full-full situation), there was flow connecting those two detectors. Even though that was usually the case, it is part of the uncertainty in our data. In order to reduce the possibility of false negatives (i.e., an empty OFD when there was overland flow [3]), we tested the OFDs in the laboratory and took the following precautions in every campaign: The OFDs were installed on the lowest part of the channel or flow line; after every observation, the OFD was repositioned appropriately; the OFD was considered active when it was at least half full; and we tried to gather data from as many events as possible. Besides taking those precaution measures, field notes, regarding if the stream reaches were still wet or if there was any litter and sediment dragged or deposited in the flow lines, were taken in every campaign

We recorded each OFD and intermittent segment location using a GPS receiver with typical accuracy of 7 m or better. Due to the inaccuracy of the GPS data in relation to the extension of the study area, the drainage length was measured manually with a measuring tape. The channel extension was defined up to the place where the bed channel was no longer visible and there were no signs of flow. We monitored the occurrence of overland flow during rainfall events in ephemeral flow lines from July 2018 to October 2019. We checked each OFD response no earlier than 2 h after the end of a rainfall event [3]. In the period, we recorded 48 events, in time intervals that varied from two to 29 days.

*2.3. Groundwater Levels and Soil Characteristics*

Eleven wells were installed close to the flow lines in order to measure the groundwater level (Figure 1c). They were installed with a manual auger and PVC pipes of 5-cm diameter, the lower ends of the pipes were grooved and covered by a geotextile blanket. The depth of the well was adopted as the depth of the soil and varied from 0.73 m to 4.10 m.

Inside each well there was a rod containing reservoirs at known heights that filled with water as the groundwater level increased. The upper-most reservoir that was filled was considered to mark the

maximum water level for the event, with an error equal to the distance of two consecutive reservoirs. Groundwater level was also measured manually on every campaign. The groundwater level was considered as the distance from the water to the land surface. When there was no water in any reservoir or when the well was dry, an absent value was registered. We cannot say that there was no water because the level could be below the level of the first reservoir or below the lower limit of the well. Due to rock outcrops and very heterogeneous soil, we dug several holes around W11 and we used it for the soil depth information only.

Tests for the determination of saturated hydraulic conductivity ($K_s$) were carried out in five locations (Figure 1c) using the inverse auger hole method as described by Ojha et al. [38] and the procedures recommended by the Brazilian Association of Engineering and Environmental Geology [39]. The infiltration capacity was estimated in six locations using a double-ring infiltrometer (Figure 1c). We selected a varied range of locations and took care to keep the vegetation cover intact. Table 1 shows minimum, median, and maximum values for all tests conducted in the catchment. The saturated hydraulic conductivity decreased with the depth of the soil. On the surface, the infiltration capacity was several times greater than the $K_s$ of the lower layers.

Granulometry analysis was carried out on hillslope and streambed soil samples (Figure 1c). The sampling depth was chosen based on a significant visual difference in color and texture. Following the texture distribution triangle described by the United States Department of Agriculture [40], the samples showed considerable heterogeneity among the soil layers. The channel sample that was taken was mainly composed of loamy sand (77% sand). Combined with the data from that sample, we completed a visual inspection of the entire channel and sand seemed to be the dominant fraction of the soil composition. Three samples were taken on the hillslope, at 1.0 m, 1.60 m, and 2.0 m, and the corresponding granulometry was clay (66% clay), sandy clay loam (35% sand), and slay loam (47% clay), respectively. Those values are in accordance with the official soil classification maps of the area [35].

**Table 1.** Summary of infiltrability and saturated hydraulic conductivity ($K_s$). Data from the Retiro catchment.

| Depth (cm) | Infiltrability/$K_s$ (mm·h$^{-1}$) | | | Sample Locations |
|:---:|:---:|:---:|:---:|:---:|
| | **Min** | **Median** | **Max** | |
| 0 [a] | 120.0 | 480.0 | 1800.0 | 6 |
| 0–29 [b] | 4.11 | 10.14 | 28.01 | 5 |
| 29–90 [b] | 1.45 | 2.42 | 16.66 | 5 |

[a] Infiltrability measured with a double-ring infiltrometer. [b] $K_s$ measured with inverse auger hole method.

## 2.4. Rainfall Events

Rainfall was measured using a tipping bucket-recording rain gauge located 200 m from the study area. The data logger recorded the time of each 0.2-mm tip. In order to analyze rainfall intensity per minute, we counted the number of tips in 1-min intervals. Forty-eight precipitation events with at least 1 mm separated by dry intervals greater than 24 h were identified to ensure that there was no overlap of events. If there was more than one event in those conditions between two monitoring campaigns, it was assumed that the event with the highest total precipitation was the one that generated the overland flow. The maximum number of events in one observation period was four.

For each event, the total rainfall, duration, maximum intensity of precipitation in 5 and 60 min were calculated, as well as the number of times that the average saturated hydraulic conductivity ($K_s$) of the top and bottom soil layers was overcome.

The antecedent precipitation index (API, in mm) was calculated for each storm event using Equation (1), as proposed by Kohler and Linsley [41] as an estimation of soil wetness:

$$API = b_1 P_1 + b_2 P_2 + b_3 P_3 + \ldots + b_i P_i \tag{1}$$

where $P_i$ is the total precipitation which occurred $i$ days prior to the storm under consideration and $b_i$ is a constant, which is assumed to be a function of time as $b_i = 1/i$. We calculated API from 1 to 20 days. Throughout the text, only the value of API3 is shown, which was the only one that was related to the dynamics of the streamflow.

### 2.5. Drainage Network Extension and Connectivity

The relative length of the total active drainage network (ADN) was determined by summing the lengths of stream segments' contribution of each active overland-flow detector for each observation period. The connected drainage network (CDN) was determined by summing the lengths of all stream segments whose flow reached the outlet without crossing any empty overland-flow detector (e.g., [17]). Figure 2 shows typical situations for calculating ADN and CDN, in which the stream flow is connected to the outlet (Case 1): The flow occurred in part of the channels but did not reach the outlet of the watershed (Case 2) and the flow was fragmented and only part of the active channels were connected (Case 3). In all cases, the sum was divided by the total extent of the drainage network.

The extension of the active drainage network and the extension of the connected drainage network were correlated with the characteristics of the rainfall events (predictor variables) through the Spearman coefficient ($\rho$) and the coefficient of determination ($R^2$).

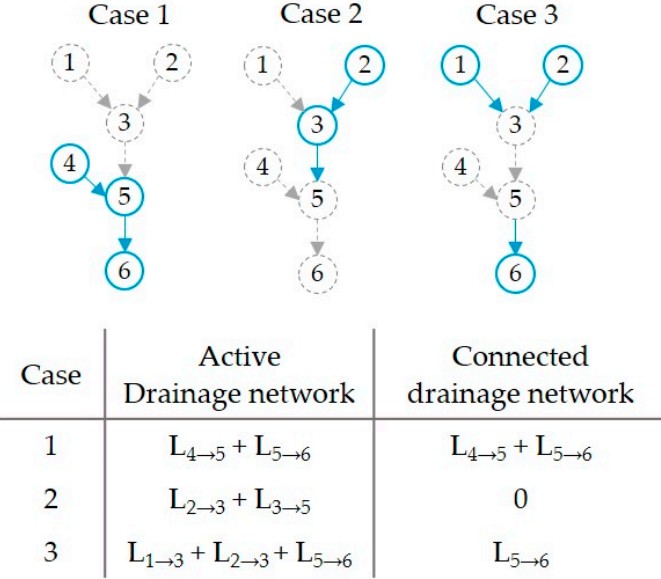

| Case | Active Drainage network | Connected drainage network |
|:---:|:---:|:---:|
| 1 | $L_{4\rightarrow5} + L_{5\rightarrow6}$ | $L_{4\rightarrow5} + L_{5\rightarrow6}$ |
| 2 | $L_{2\rightarrow3} + L_{3\rightarrow5}$ | 0 |
| 3 | $L_{1\rightarrow3} + L_{2\rightarrow3} + L_{5\rightarrow6}$ | $L_{5\rightarrow6}$ |

**Figure 2.** Metrics for the active drainage network and connected drainage network in three possible situations. Case 1, part of the river connects to the outlet. Case 2, there is flow in the channels, but it does not reach the outlet. Case 3, parts of the channels are disconnected. The numbered circles represent the overland-flow detectors. Continuous circles and arrows represent the OFDs and channels that had water. Dashed circles and arrows are the dry channels and OFDs for the respective case. The last circle represents the drainage outlet. The table shows the logic of the calculations. L is length.

## 3. Results

### 3.1. Overland-Flow Occurrence

We observed flow in at least one stream segment in the network channel in 44 of the 48 events. The total extension of the active drainage network varied between 0 to 97%. The intermittent reach was active in 36 events. In four events there was streamflow in the intermittent channel only. The drainage network was connected to the outlet in 21 events. The CDN remained below the ADN in 16 of these events, showing that the flow occurred in the channels, but part of them did not connect to the outlet (e.g., Figure 3c,d). In five events, the CDN was equal to the ADN, as the entire slope was connected

to the outlet (Figure 3f). When the connection was reached, the connected network extension values ranged from 15 to 97% of the total drainage network, while the active network extension ranged from 4 to 97%. The antecedent wetness did not strongly influence on this dynamic.

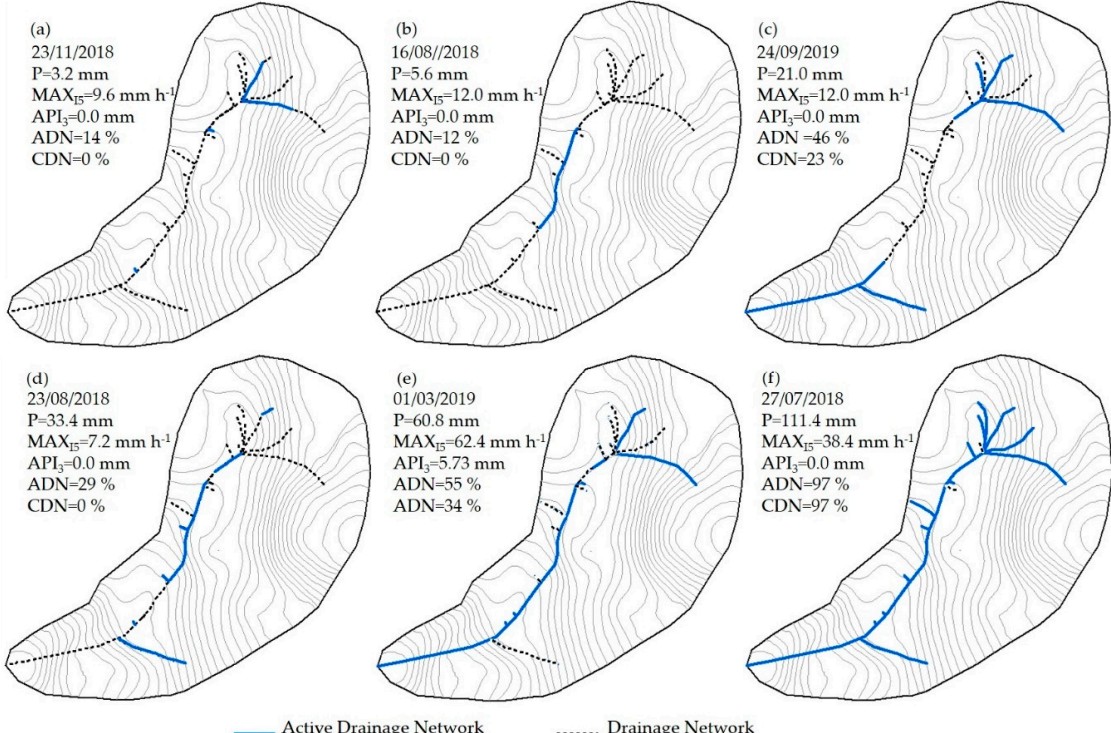

**Figure 3.** Total length of the active drainage network for different precipitation events. (**a**) Low active drainage network. (**b**) Only the intermittent section was active. (**c**) Ephemeral flow in different locations. (**d**) High ADN but no connection. (**e**) High precipitation intensity with inactive channels. (**f**) Maximum AND and CDN. Events are organized from lowest to highest total precipitation. For each event, total precipitation (P), maximum 5-min precipitation intensity ($MAX_{I5}$), $API_3$, active drainage network (ADN), and connected drainage network (CDN) are shown.

The length of the active drainage network increased with increasing total precipitation ($R^2 = 0.65$, $\rho = 0.78$) (Figure 4a). Four classes of events were created to determine those thresholds using the total precipitation and the connected drainage network. The first class is the events in which there was no connection and the ADN was low. In the second class, we have high-precipitation events in which runoff occurred, but it was not connected to the outlet. In the third class we have the events in which connection occurred, but part of the channels was disconnected. Finally, in the fourth class we have the events in which the CDN approached or equaled the ADN, and the entire hillslope became connected, reaching maximum efficiency in transmitting water as surface flow.

There were events that the drainage network connected to the outlet starting from 10 mm of total precipitation. Therefore, we set the 10-mm threshold for Class 1. Class 2 and Class 3 events are in the same total precipitation range (10–80 mm) but with different CDN, so we determined the threshold of 35% active drainage network to separate Class 2 from Class 3 events. That separation allowed the exploration of other controls on CDN apart from total precipitation in those cases. The extension of the connected drainage network increased linearly with the extension of the active drainage network ($R^2 = 0.85$, $\rho = 0.88$). In some cases, there were events with similar total rainfall but different responses. For example, in events between 30 and 40 mm, the ADN ranged from about 20% to 70%, while the connected drainage values ranged between 0 and 44%. Events above the 85-mm threshold were classified as Class 4. They were very distinct, as ADN and CDN values were close to 100%.

Total precipitation exerted the greatest influence on the extent of the active network and connectivity as is evidenced by the Spearman's correlation coefficient (Figure 4b). There were also high correlations with maximum short-term intensities, while short-term antecedent moisture indices had a low correlation with runoff occurrence. The frequency at which the maximum precipitation intensity exceeded the hydraulic conductivity of the soil in the upper and lower layers of the soil was of some predictive power, since $K_s$ decreased rapidly in the shallow soil layers of this hillslope.

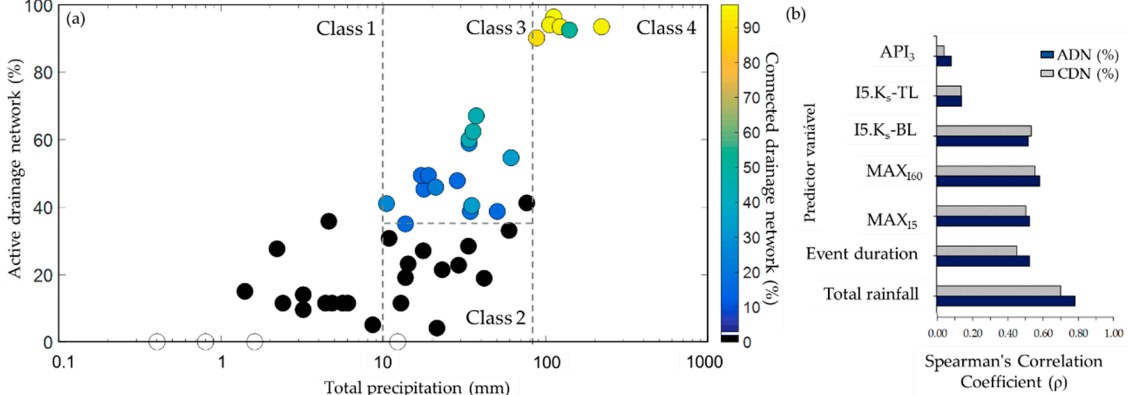

**Figure 4.** (**a**) Active drainage network and total precipitation. The color scale indicates connected drainage network (%). The unfilled circles are events in which overland flow was not detected. The dashed lines show the separation of events into four classes. (**b**) Spearman's correlation (ρ) for % connected drainage and % active drainage. The variables are: $MAX_{I5}$ and $MAX_{I60}$ (maximum 5- and 60-min precipitation intensity), $I5.K_s$-BL (number of times that the maximum intensity of 5min exceeded the average $K_s$ of the lower layer of the soil), $I5.K_s$-TL (number of times that the maximum intensity of 5 min exceeded the average $K_s$ of the topsoil), and $API_3$ (antecedent precipitation index of three days).

### 3.2. Development of Hillslope Connectivity

All predictive meteorological variables were compared for the four classes of events. Classes 2 and 3 were in the same range of total precipitation values, but the network connected to the outlet only in the events of Class 3 (Figure 5a). The three-day antecedent precipitation index ($API_3$, Figure 5b) and the maximum 5-min precipitation intensity (Figure 5c) were the only predictor variables that were different for Classes 2 and 3, with a difference in the mean and position of the second and third quartiles. Class 3 events showed the greatest dispersion in terms of antecedent wetness but were concentrated in low values. The contribution area of the active overland-flow detectors was also evaluated for the four classes (Figure 5d). There was an area threshold for the start of the connection of 0.5 ha (corresponding to 20% of the total area of the basin) which was only reached in Class 3.

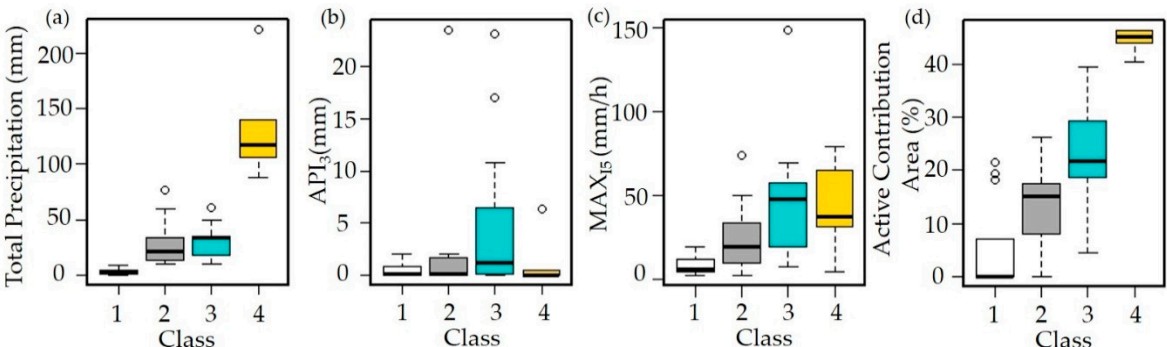

**Figure 5.** Boxplot of predictor variables for the four event classes. (**a**) Total precipitation, (**b**) antecedent precipitation index ($API_3$). (**c**) Maximum 5minute precipitation intensity ($MAX_{I5}$). (**d**) Active contribution area. Open circles are outlier values.

The percentage of active area varied among events (Figure 6), with only four locations contributing to runoff in Class 1 events (Figure 6a). There was a great difference in the frequency of activation of each OFD in Class 2 and Class 3 (Figure 6b,c). The areas with the greatest change were concentrated in the high slope region, and their slopes are in the reforested and more altered region. For example, one of the OFDs was active in 33% of Class 2 events while it was active in 87% of Class 3 events. Another OFD activation went up from 17% to 93% in Class 3. In Class 4, virtually the entire slope contributed to runoff (Figure 6d). Two OFDs were never active.

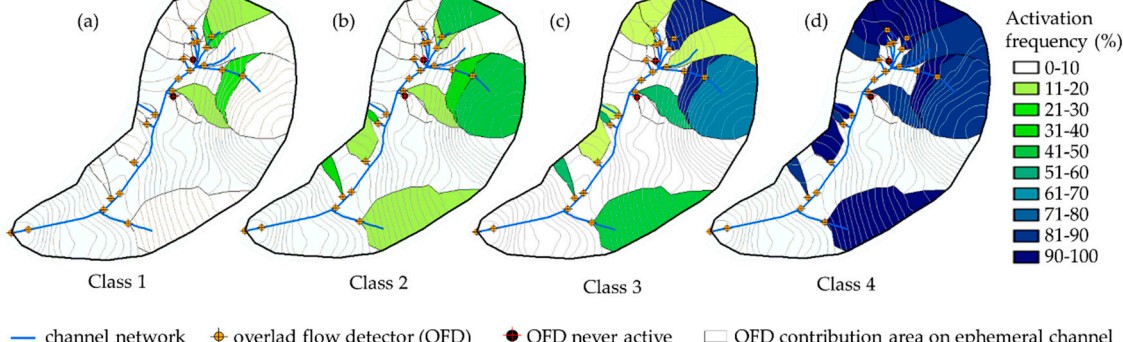

**Figure 6.** Frequency of activation of the contribution areas for each event class. (**a**) Class 1. (**b**) Class 2. (**c**) Class 3. (**d**) Class 4.

### 3.3. Topographic Controls

The frequency of activation of OFDs was weakly related to the topography attributes that we evaluated (Figure 7). There did not seem to be any relationship between OFDs' activation and channel slope (Figure 7b), upslope area (Figure 7a), topographic wetness index (TWI) (Figure 7c), drainage density (Figure 7d), and convex and linear curvature concavity (Figure 7g,h). Although weak, there seemed to be a decrease in the frequency of activation with increasing concavity (Figure 7f). Accentuated slopes led to a higher frequency of activation (Figure 7e).

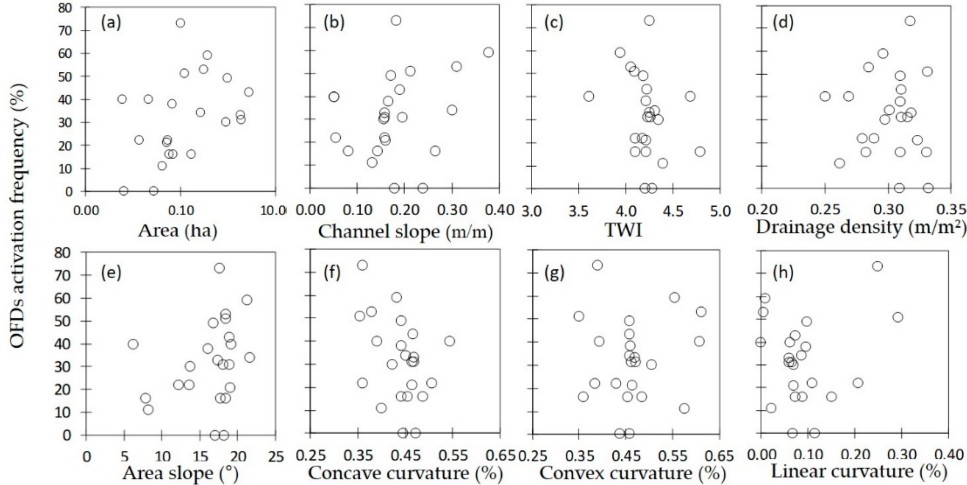

**Figure 7.** Overland-flow activation frequency in relation to topographic indices. (**a**) Area. (**b**) Channel slope. (**c**) Topographic Wetness Index. (**d**) Drainage Density. (**e**) Area Slope. (**f**) Concave Curvature. (**g**) Convex Curvature. (**h**) Linear curvature. Each circle represents one of the 23 overland-flow detectors.

### 3.4. Overland-Flow Response to Water Table Development

When we look at the entire slope (Figure 8), we see that the flow occurred mainly in the intermittent section. In 36 of the monitored events, water was flowing in this location. This runoff stability was

verified in field surveys when streamflow was persistent even after days without rain. In Class 2 and Class 3 events (Figure 8b,c), the number of active channels and their frequency of activation increased. Note that the activation frequency of the reach between wells W4 and W5 and wells W7 and W8 was less than that of the upstream channels in the Class 2 events. This also happened for wells W3 and W5 and W7 and W8 in the Class 3 events. These low frequency activation sites were the main sites where disconnection occurs. In addition to being the transition sections between ephemeral and intermittent flows, they are places prone to sediment (i.e., mainly sand) deposition in the streambed.

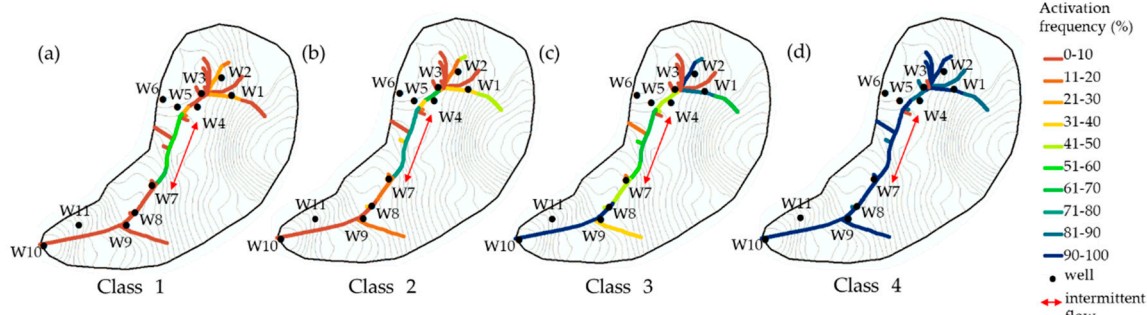

**Figure 8.** Channel activation frequency for each event class and location of each well. (**a**) Class 1. (**b**) Class 2. (**c**) Class 3. (**d**) Class 4. Each separate colored channel section indicate the position of an overland-flow detector. The arrow indicates the length of intermittent flow section.

We estimated the depth of the soil along the stream using the depth of the wells next to the channels (Figure 9a). The hillslope's soil depth varies along the main channel. Where the soil is deep, between wells W2 to W5 and wells W7 to W11, the channel segments are ephemeral. In the middle region of the basin, between W5 to W7, the intermittent runoff coincides with the shallow soil. Between those wells, the slope of the main channel is relatively steep compared to its downstream and upstream reach. The average slope of the drainage network changes from 0.06 m/m between W2 and W5 to and 0.2 m/m between W5 and W7, and changes again to 0.04 m/m between W7 and W11. In the lower region of the basin, between W11 and W10, the soil is shallow, the channel bed is covered by boulders, and the flow is ephemeral. The average slope in the lower channel is 0.76 m/m. We can see that the water level in wells W1, W2, and W3, in the upper part of the basin, is always below 1 m depth (Figure 9b). Even though the level in those areas is below that of the ephemeral channels, OFDs in the channels near those wells are among the most active in the basin.

At the beginning and at the end of the intermittent region, around wells W5 and W7, the groundwater level reached values close to the surface. The water level in well W7 reached the surface level for one event, with the shallow layers saturated. Those wells were dry or at their lowest level during the same events that the flow ceased in the intermittent channel reach. Between W7 and W11 the depth of the ground increases abruptly, reaching 4 m. The slope becomes ephemeral again, and the groundwater level is close to 1 m or above for most events.

Since the soil around the outlet well (i.e., W10) is shallow, it quickly reaches levels close to the surface. Although it would be intuitive that water levels in this well are related to the connection of the outlet, with groundwater contributing to the channel, there is no relationship between the activation of OFD in the outlet and the level in that well. The connection between the slope and the outlet occurred even in events of high total precipitation for which the well W10 remained dry. On the other hand, the connection did not develop when the well reached levels close to the surface (0.2 m) for high-precipitation events (77 mm and 60 mm).

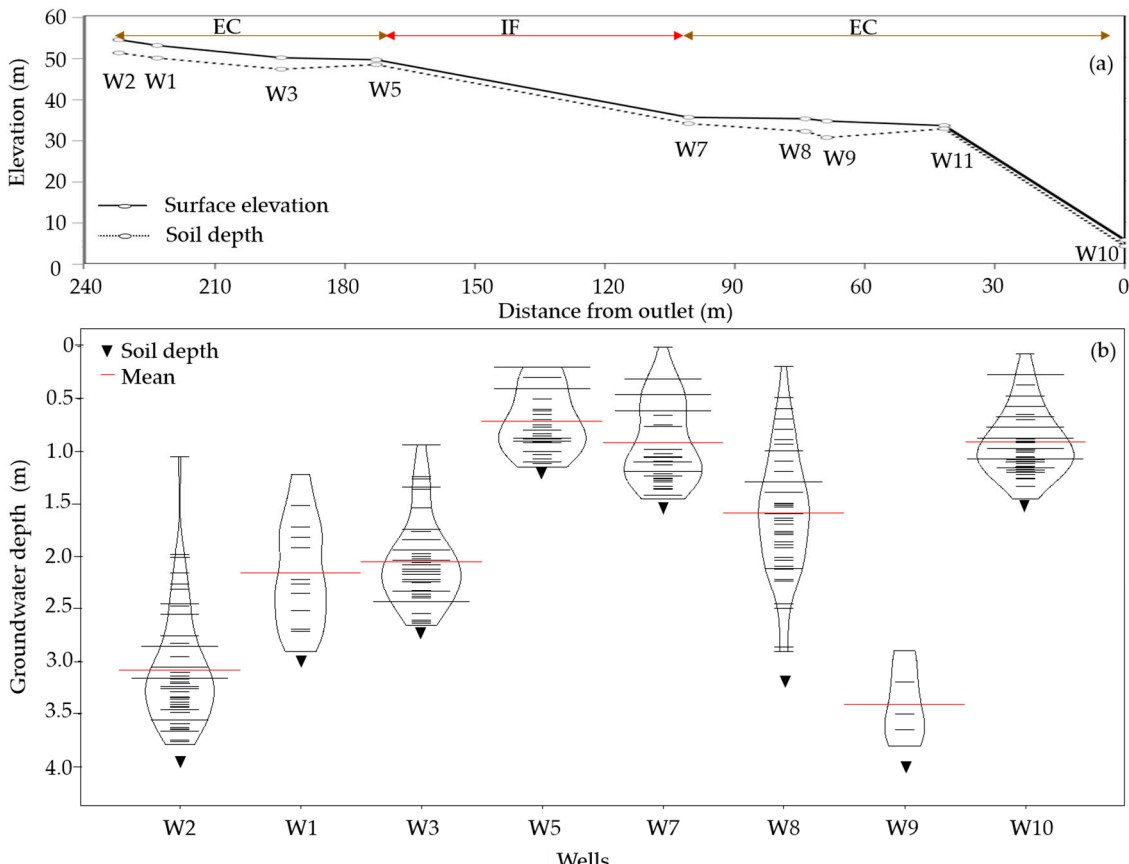

**Figure 9.** (**a**) Longitudinal profile of the main channel and measured soil depth. The ephemeral (EC) and intermittent flow (IF) parts are highlighted. The solid line connecting the wells is for illustration purposes as the wells follow the slope but are not located exactly in the channel. (**b**) Measured and maximum event groundwater levels (the y-axis is not in scale with the topography). The polygon shape in the bean plot is given by a normal density distribution. Each black line is a measured data. The width of a line is proportional to the measurement count. Wells W4, W6, and W11 are not shown. The black triangles indicate the approximate depth to the bedrock.

## 4. Discussion

In humid, forested areas with shallow soils, the flow in temporary channels can occur as an expansion of the perennial network towards the headwaters [3]. In low relief landscapes with deep soils, the unpredictable extension variation of the drainage network was documented, occurring due to the outcrop of the water table [20]. We found that the source areas producing flow can be highly variable, similar to other studies in very different environments (e.g., Mediterranean climate mountains, USA [17]; peatland, UK [8]; subtropical humid forest, USA [20]; humid temperate climate, Switzerland [42]; semiarid Caatinga, Brazil [43]). In our case, we attribute this behavior to the strong variation in soil depth, which alters the distribution of moisture and storage capacity in the basin.

The deep soil keeps the groundwater away from the surface, causing runoff to cease quickly after events. It may sound counterintuitive that water flow will cease in deep soils with higher storage capacity, however, at this scale we are mainly seeing water stored in the soil which is not necessarily coming from deep groundwater storage. Therefore, besides soil depth, flow permanence might be controlled by the interaction of topographic characteristics (e.g., slope and contribution area) and soil properties. Where the soil is shallow and steep, between W5 and W7, the water table emerges on the surface, fed by the deep and gentle soils upstream, maintaining the flow beyond the time scale of the event. The channel morphology must also be considered in the runoff generation. The intermittent channel has a well-defined channel bed and is mainly composed of sand. In the lower part of the slope,

near the wells W11 and W10, the soil is also shallow and steep, but it is formed by boulders, which could increase the transmissivity of the soil and consequently increase the losses through infiltration.

The total precipitation threshold for the beginning of runoff is very low and almost any event can start the runoff in the slopes. The density of active drainage increases with total precipitation, but the places of origin of the flow can be highly variable. The fragmentation of the flowing network leads to a high density of active drainage, but a low density of connected drainage. The antecedent moisture control over the beginning of the hillslope-outlet connection is clear for events larger than 10 mm. If the antecedent moisture of the basin is high and it does not rain with enough maximum intensity to overcome the transmissivity of the soil, the connection will not be reached. When connectivity is established, it increases with increasing total precipitation. There is a limited flow capacity and the moisture threshold must be exceeded long enough for the process to have a significant impact, connecting the slope to the channels as it has been noticed in other studies [18,19]. During high-intensity events there may be an increase in runoff even though the soil moisture threshold has not been exceeded. For Class 4 events (>85 mm), total precipitation and intensity control drainage density and connectivity. The study basin shows a mixture of processes dependent on intensity and on moisture for establishing the connection of the slope by surface paths, something that has rarely been reported for humid catchments [18].

Variations in flow distribution and flow travel time are largely controlled by the topography and infiltration capacity of the soil [27]. The high rate of soil infiltration and the low saturated hydraulic conductivity (Table 1) can provide soil saturation close to the surface, suggesting the occurrence of saturation-excess overland flow [3]. The direct relationship between the frequency of OFDs' activation and the slope area [27] and the inverse relationship between frequency of OFDs' activation and the concavity of the contribution areas [44] is an indication of fast flow, facilitated by the topographic characteristics despite the relationship being weak. The most active OFD was located where the contribution area changed the most. The difference in the frequency of activation of the sensors can be explained by the anisotropy of soil properties [45] and vegetation characteristics [43]. Soil structure and saturated hydraulic conductivity can be changed in areas with different vegetation stages [46], making some regions more conducive to flow generation [16].

The influence of vegetation has not been quantified; however, the high infiltration rates are the first indication that the structure of the forest has a significant role in the generation of runoff, mainly in processes that occur close to the surface. The discontinuity of the flow seems to depend on the balance between the losses that occur along the channel and the availability of water upstream. Quantifying infiltration losses in the channel and evaporation could help to predict the sites susceptible to the flow disconnection and the impact on the water availability in those environments. This type of information is important mainly in basins that are used for water supply, where the quality and quantity of available water has the greatest social and environmental impact.

## 5. Conclusions

We used rainfall data and overland-flow detectors to identify the dynamics of the drainage network and controls over connectivity in a steep ephemeral-intermittent slope, with a strong variation in soil depth. Our main findings can be summarized as follows:

1.  Even low total precipitation is enough to generate runoff. The extent of active drainage increases directly with the size of precipitation events. The rapid decrease in the hydraulic conductivity of the soil close to the surface is an indication that runoff may start with a suspended and temporary water table. The spatial distribution of ADN does not follow a pattern and can occur in a fragmented way. On the other hand, the most active steep slopes are those with the most altered vegetation cover and with a large cumulative area slope.

2.  The slope-outlet connection develops when the thresholds for total precipitation (>10 mm) and active contribution area (>20%) are exceeded. The area threshold depends on the fitting combination between antecedent moisture and maximum 5-min rainfall intensity. There is an

upper total precipitation threshold (>85 mm) in which maximum connectivity is achieved and the other characteristics of the event have no influence.

3. The change in runoff permanence reflects the variation in soil depth. In deep soils the channels are ephemeral and where the soil is shallow the groundwater contributes to the channel making the flow intermittent. The connection efficiency of the slope is affected by the infiltration of the flow in the channel where the soil changes from deep to shallow.

**Author Contributions:** Conceptualization, A.B.A.P. and P.L.B.C.; methodology, A.B.A.P.; formal analysis, A.B.A.P.; investigation, A.B.A.P., C.I.d.S., P.F.A. and J.H.M.S.; data curation, A.B.A.P.; writing—original draft preparation, A.B.A.P.; writing—review and editing, A.B.A.P. and P.L.B.C. All authors have read and agreed to the published version of the manuscript.

**Funding:** FAPESC (in Portuguese, Fundação de Amparo à Pesquisa e Inovação do Estado de Santa Catarina) provided the scholarship for the first author, CAPES (in Portuguese, Coordenação de Aperfeiçoamento de Pessoal de Nível Superior) provided the scholarship for the second authors, and CNPq (in Portuguese, Conselho Nacional de Desenvolvimento Científico e Tecnológico) provided the scholarship for the third and fourth authors. Part of this research was funded by CNPq by process MCTI/CNPQ/Universal 14/2014 and MCTIC/CNPq No 28/2018.

**Acknowledgments:** The authors acknowledge the Fundação Municipal do Meio Ambiente (FLORAM) team for its support in field expeditions and installation of equipment and the team at Casa de Retiros Vila de Fatima for allowing the use of its space and access to the study area. Thanks to Kathy and Ken Babiuk for proofreading the article. The authors appreciate the support of Guest Editor Stephanie Kampf.

**Conflicts of Interest:** The authors declare no conflict of interest.

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
