# Peer review of "Connectivity of Ephemeral and Intermittent Streams in a Subtropical Atlantic Forest Headwater Catchment"

_water, doi:10.3390/w12061526_

Round 1

Reviewer 1 Report

See attached.

Author Response

Response to Reviewer 1 Comments

The authors gratefully appreciate the helpful comments of Reviewer #1 that greatly improved our manuscript. Please, kindly find below our reply to all the comments. Answers are also attached.

Reply to Major comments

Major Comment #1: I believe the groundwater data is underutilized in this analysis. It is difficult to connect the different flow events to the water level dynamics. Thus, I recommended some alternative ways to present the groundwater data in comments below.

Reply: We agree that we have not fully explored the groundwater data in the analysis and that it is difficult to make a direct connection with the overland flow detection. We have followed the reviewer’s suggestions in trying to present the groundwater in an improved way. Please find below the reply to the specific comment SC#18.

Major Comment #2: L 374-378 This result seems counterintuitive. It is typically thought that watersheds with deeper soils have higher storage capacity and thus can sustain baseflow for longer. Here you are showing the opposite. Is this suggesting that the channel is generally losing (not gaining water)? I think the authors have an opportunity to build on this interesting and possibly counterintuitive finding more.

Reply: The reviewer is right; our results suggest that the channel is generally losing water. We agree that it is counterintuitive and that might be partly related to the scale we are analyzing. We believe that at this scale we are mainly seeing water stored in the soil which is not necessarily coming from deep groundwater storage. Therefore, besides soil depth, flow permanence might be controlled by the interaction of topographic characteristics (e.g. slope and contribution area) and soil properties. We have rewritten that part of the manuscript in order to make it clearer and build on that finding (L369-374 of the revised manuscript).

Major Comment #3: The manuscript in general needs editorial work there are many issues with grammar and sentence structure.

Reply: We have edited the manuscript and revised the grammar and sentences structures as required.

Reply to Specific comments

Point 1: L 12 the first sentence feels like it is missing a world. Specifically, what thresholds are the authors ‘referring to? Moisture thresholds?

Response 1: We have rewritten that sentence as:

“Stream network extension and contraction depend on landscape features and the characteristics of precipitation events.”

Point 2: L 44-50 These sentences are confusing and not clear. From how they are written, I cannot tell what is means by “a gains and loss system that depends on the transmissivity characteristics of the channel”. Does this mean that different portions of the stream channel can change from losing to gaining reaches? Similarly the next few sentences are confusing and could use some editing to make more clear.

Response 2: Those sentences were confusing indeed. Yes, we meant that different portions of the stream channel could change from losing to gaining reaches. We have rewritten those sentences as follows (L44-50 of the revised manuscript):

“Different portions of the stream channel can change from losing to gaining reaches affecting the extension and connection of the drainage network. Part of those dynamics are controlled by the transmissivity characteristics of the channel [17]. It is possible that a low-intensity rain will generate runoff, but water will be retained or infiltrate in the channel due to local variation in transmissivity. On the other hand, if the rain occurs long enough, the amount of water available will overcome the moisture deficit and the stream will be connected along the slope [18,19].”

Point 3: L 51 This threshold for runoff generation is true for some environments, but not all. I would encourage the authors to edit this sentence to be clearer about this, otherwise it sounds like this is the only way that runoff generation can occur. It seems more inclusive to state something about how there are many mechanisms, depending on the climate, lithology, land use, etc. of the watershed, for streamflow activation to occur.

Response 3: We agree that those sentences sounded as the only way that runoff generation could occur. We tried to be more inclusive and have rewritten that part as follows (L52-64 of the revised manuscript):

“The dynamic of temporary flows depends on the complex interplay of the landscape attributes (e.g. slope, soil depth, lithology, land use) and the characteristics of the precipitation events. While in some environments the temporary flow results from the expansion of the perennial drainage network [3], in other areas channel network expansion and connection may be highly variable. The activation of the flow path in Mediterranean climate mountains for example [16] was attributed to the interaction between precipitation and the great variation in the saturated hydraulic conductivity (Ks) of the soil. Where Ks gradually decreases, deeper water tables are formed. The fast decrease of Ks in the superficial layers of soil may promote the formation of a shallow, suspended and temporary water table [18]. The same characteristic has been observed in other highly seasonal humid climate basins (e.g. a tropical lowland forest in Panama [3], a humid subtropical pine forest in USA [20] and a grassland area in Australia [18]). In a subtropical humid forested hillslope, the beginning of runoff was related to the total rainfall threshold necessary to fill the depressions of the soil-bedrock interface [21,22].”

Point 4: L 51 58 This paragraph would benefit from some discussion about why some watersheds are driven by different runoff generation mechanisms. That is, why is there so much variability in what causes runoff to activate in the examples given? Where are these examples located in the world? Are they all from steep, forested watersheds? Are they from temperate, humid, arid landscapes?

Response 4: We have improved the discussion in that paragraph. Please, see the reply to SC#03 above.

Point 5: L 69 what does “timidly explored” mean?

Response 5: We have deleted that expression. L75 of the revised manuscript.

Point 6: Table 1 It is not clear immediately clear that the depth 0 is associated with infiltrability and the 0-29 and 29-90 are Ks measurements. Can the authors make this more clear? Also are these samples taken from the same location, or 5-6 different locations? I cannot find any information where these measurements are taken. Are they in the streambed or hillslope? This would influence how these values are interpreted.

Response 6: We have modified the Table description in order to make it clearer. The samples were taken from different locations, so we have modified where it said “sample size” to number of “sample locations”. We have also included in Figure 1 the places where those samples were taken from.

Point 7: L 155-160 Similar to my concerns about the lack of information of the infiltration and Ks information, it is not clear where the soil texture data is collected. Can you place the sample locations on the Figure 1 site map? I suspect there are differences in soil properties and thus infiltration/Ks values in the secondary forest and reforested area of the watershed, thus having information on where measurements were taken would be helpful for interpretation of the data. Further, how many samples were taken in the channel were there enough taken to feel confident that the channel is composed of loamy sand?

Response 7: We have placed the sample locations on the Figure 1 site map. We have not found differences in soil properties in the secondary and reforested area.

We took one sample in the channel and we agree that it is not an exhaustive sampling campaign in order to be completely confident of the soil texture in the full extension of the channel. However, we have done a visual inspection of the entire channel and we believe that sand is the dominant fraction of the soil composition. We have also compared the soil samples with the official soil map of the area (provided by Embrapa, 2004) and they are compatible. We have tried to make it clearer in L185-190 of the revised manuscript.

Point 8: L 166 Did the tipping bucket record the time for each 0.20 mm tip, or did it record how many tips per a certain amount of time? If it recorded number of tips within a certain time interval, what interval was that?

Response 8: The tipping bucket recorded the time of each 0.20 mm tip. The datalloger did not register the number of tips within a certain time interval. In order to analyze rainfall intensity per minute, we counted the number of tips in 1-minute intervals. (L195-197 of the revised manuscript)

Point 9: L 175 Most readers (including this reviewer) is not familiar with all the API methods out there. Thus, it would be helpful to provide justification for the Kohler and Linsley method with at least one sentence explanation of how this API is calculated.

Response 9: We have included the equation we used to calculate API. Please, check L205-209 of the revised manuscript.

Ponit10: L 179-186 Is it possible that there is not continuous flow between the overland flow detectors? I can’t actually find information in the text about what the spacing is of the detectors. Further, it does not appear there are flow detectors in the intermittent stream section on the site map. Is that because there was always flow on that portion of the channel? Do the authors feel that the overland flow detectors provide enough information to feel confident if there is flow at two detectors on the same tributary then that means there is flow connecting the two detectors? If so, the authors need to state this.

Response 10: Yes, it is possible that the flow is not continuous between two overland flow detectors. We have adopted similar assumptions to Zimmerman et al (2014): if two adjacent OFD are full (full-full situation) we considered continuous flow between them; if one OFD is active and the adjacent downstream one is empty (full-empty situation) we considered that the flow occurred up to the first OFD; if an OFD is empty and the adjacent downstream one is full (empty-full situation) we considered that flow started from the downstream full OFD. Please, see L140-146 of the revised manuscript.

            We could not install OFDs using uniform spacing between them. The channel morphology is heterogeneous, and we could not install the OFD anywhere we wanted. For example, due to cobbles and boulders in the channel there were only two spots where we could install OFDs in the reach between W8 and W10. We have even seen water flowing beneath those boulders in a few occasions. We installed the OFDs where the channel characteristics allowed and avoiding big distances among them. Please, see L130-133 of the revised manuscript.

            The reviewer is right, there are no OFDs in the intermittent stream section. In every campaign we took note on the conditions of the channel reach and there was water flowing in that section in most of the time. Please, see L136-138 of the revised manuscript.

            We had to make the assumption that if two adjacent OFD were full (full-full situation) there was flow connecting those two detectors. We expect that is the case in most of the times, however we have to accept that as part of the uncertainty in our data. It is also possible that we registered false negatives (an empty OFD when there was overland flow) as detailed by Zimmerman et al (2014). In order to reduce that possibility of false negatives, we tested the OFDs in the laboratory and also took the following precaution in every campaign: the OFDs were installed on the lowest part of the channel or flow line; after every observation, the OFD was repositioned appropriately; the OFD was considered active when it was at least half full; we tried to gather data from as many events as it was possible. Besides taking precaution measures with the OFDs, we took field notes in every campaign. For example, we wrote if the stream reaches were still wet or if there was any litter dragged and deposited in the flow lines. Please, see L144-152 of the revised manuscript.

Point 11: 194-202 How did the authors determine the quantitative thresholds for the different classes? For example, how did the authors choose the threshold between class 1 and 2?

Response 11: We have visually inspected Figure 4 in order to determine those thresholds using mainly the Total Precipitation and the Connected Drainage Network (L251 of the revised manuscript).

We can see that there are events that the drainage network connects to the outlet starting from 10 mm of Total Precipitation. Therefore, we set the 10mm threshold for Class 1. As Class 2 and Class 3 events are in the same Total Precipitation range (10-80mm) but with different CDN, we thought that we could learn what was controlling CDN apart from Total Precipitation in those cases. So we determine the threshold of 35% Active Drainage Network to separate Class 2 from Class 3 events. Class 4 events are very distinct with the ADN and CDN close to 100%. We have improved this description in L254-263 of the revised manuscript.

Point 12: L 205-206 Why say 37 of 51 events produce ephemeral flow and 73% have intermittent flow. 73% of 51 equals 37 events as well. What is the difference between these observations?

Response 12: Percentage values were exchanged for the number of events. We tried to be more clear and have rewritten that part as follows (L236-244 of the revised manuscript):

“We observed flow in at least one stream segment in the network channel in 44 of the 48 events. The total extension of the active drainage network varied between 0 to 97 %. The intermittent reach was active in 36 events. In 4 events there was streamflow in the intermittent channel only. The drainage network was connected to the outlet in 21 events. The CDN remained below the ADN in 16 of these events, showing that the flow occurred in the channels, but part of them did not connect to the outlet (e.g. Figure 3c, d). In 5 events, the CDN was equal to the ADN, as the entire slope was connected to the outlet (Figure 3f). When the connection was reached, the connected network extension values ranged from 15 to 97% of the total drainage network, while the active network extension ranged from 4 to 97%. The antecedent wetness did not strongly influence on this dynamic.”

Point 13: L 208 Which two OFD were never active?

Response 13: We have highlighted them in Figure 6 of the revised manuscript.

Point 14: Figure 3. If the API3 is zero for all of these events, why even report it? Why not report the API for a different length of time that might highlight differences in the antecedent storage, assuming these events occurred during different wetness conditions in the subsurface. It would be also really nice to know what parts of the stream network were active before these events. That is, was the network entirely dry before the event?

Response 14: The value of API3 on 03/01/2019 was updated in Figure 3. Our intention with this figure was to show how the extent and location of the flow source is variable among events and how storage did not strongly influence on this dynamic. (L 244 of the revised manuscript)

It is not possible to know if the slope was dry before the event with non-automatic data. The ephemeral channels dry very quickly after the rainfall event. It is only possible to compare the active network analyzed with the active network in the previous event or on the field visit day, when several days have passed. Thus, we have opted to avoid the comparison with the configuration of the active drainage network prior to the analyzed event. Please refer to L135-136 of the revised manuscript.

Point 15: Figure 4. Why do the authors not report API3, but it is shown in Figure 3? Also if API10 was the most influential API, then it seems that should be the one that is displayed in Figure 3.

Response 15: We have fixed Figure 3 (API3 for Figure 3d should have been 5.73mm). We chose to show API3 because even though its value is 0mm for 5 out of the 6 maps, API3 was the one related to the event Class separation and it is related to the events that the hillslope connected to the outlet. We have also improved Figure 4b in order to show only the rainfall characteristics used in the analysis.

Point 16: L 263-264 - change “slope to contribute to” to “slope contributed to”

Response 16: We have changed it. Please refer to L300 of the revised manuscript.

Point 17: L 278-279 It is not clear in Figure 8 what the “intermittent section is and thus I am not sure what this sentence is referring to.

Response 17: We have added a red arrow to make the intermittent section clear in Figure 8.

Point 18: Figure 9 might be more informative as a violin plot or a bean plot. This will help show more of the data, but only requires a little modification to the figure. Here is a nice resource for variations on box plots: http://datavizcatalogue.com/blog/box-plotvariations/

Response 18: Thank you for the tip on improving the boxplot. We have substituted Figure 9b for a bean plot that we think highlights the data better.

Point 19: L 315-316 missing period after sentence.

Response 19: We have added the missing period.

Point 20: L 318 320 Godsey and Kirchner (2014) highlights that there can be discontinuous streamflow across river networks, which seems similar to the authors’ finding, though the authors say that the discontinuous nature is not similar to other studies.

Response 20: The reviewer is right and we have fixed this section accordingly (L362-367 of the revised manuscript).

“We found that the source areas producing flow can be highly variable, similar to other studies in very different environment (e.g. Mediterranean climate mountains – USA [17], Peatland – UK [8], Subtropical humid forest – USA [20], Humid Temperate climate – Switzerland [42], Semiarid Caatinga – Brazil [43]). In our case, we attribute this behavior to the strong variation in soil depth, which alters the distribution of moisture and storage capacity in the basin.”

Point 21: L 320-321 I don’t understand how the deep water table shortens the travel time of the surface flow? Could the authors explain this further and how the deep water table shortens surface flow travel times?

Response 21: The description in the text was not correct and we have improved it in the revised manuscript. We have rewritten that section as follows (L368-379 of the revised manuscript): 

“The deep soil keeps the groundwater away from the surface, causing runoff to cease quickly after events. It may sound counterintuitive that water flow will cease in deep soils with higher storage capacity, however at this scale we are mainly seeing water stored in the soil which is not necessarily coming from deep groundwater storage. Therefore, besides soil depth, flow permanence might be controlled by the interaction of topographic characteristics (e.g. slope and contribution area) and soil properties. Where the soil is shallow and deep, between W5 and W7, the water table emerges on the surface, fed by the deep and steep soils upstream, maintaining the flow beyond the time scale of the event. The channel morphology must also be considered in the runoff generation. The intermittent channel has a well-defined channel bed and is mainly composed of sand. In the lower part of the slope, near the wells W11 and W10, the soil is also shallow and steep, but it is formed by boulders, which could increase the transmissivity of the soil and consequently increase the losses through infiltration.”

Point 22: L 329- What do the authors mean by “antecedent humidity”? Hoe does humidity impact runoff generation? Similarly this word shows up in the conclusions.

Response 22: Thank you for pointing that out. We have misused the term “humidity” when we were actually referring to “moisture”. In the paper we have used Antecedent Precipitation Index (API) as a proxy for the catchment’s antecedent water storage. We have better defined the use of the API and we have used it throughout most of the text.

Point 23: L 363-364 Do the authors mean “the extent of active drainage increase directly with the size of precipitation events”?

Response 23: Yes, we have rewritten the sentence as “The extent of active drainage increases directly with the size of precipitation events.” (L418-419 of the revised manuscript).

Reviewer 2 Report

This study is interesting and I do enjoy reading it I would like to accept it for publication in Water with some minor suggestions/comments in the following.

L75-79. These sentences do not belong to Introduction.
L243. Add “of the” before “topsoil”
L282. I cannot tell that the activation frequency of the sections between wells W3-W5 is always less than the upstream channels.
L298-299. I cannot understand this sentence.
L277. Have the authors thought of hyporheic flow? Was the groundwater in the context hyporheic flow? What is the role of hyporheic flow on the occurrence of overland flow? Have the authors thought of the influence of the channel morphology and riverbed microrelief on the connectivity? The slope between W5 and W7 (Figure 9a) is relatively steep compared to its downstream and upstream reach. Could the occurrence of runoff be attributed to the steep slope rather than shallow soil?
Figure 9a. Were the wells installed in the channel? If not, I was wondering if it is proper/meaningful to draw a line to connect all the data point.
Figure 9b. No data for W11?

Author Response

Response to Reviewer 2 Comments

The authors gratefully appreciate the helpful and encouraging comments of Reviewer #2. Please, kindly find below our reply to all the comments. Answers are also attached

This study is interesting, and I do enjoy reading it I would like to accept it for publication in Water with some minor suggestions/comments in the following.

Thank you. We have addressed all your suggestions/comments below.

Reply to specific suggestions/comments

Point 1: L75-79. These sentences do not belong to Introduction.

Response 1: We understand that those sentences might seem a little unusual on the Introduction. On the instructions for authors page of Water, it is recommended that the introduction should highlight the main conclusions. We are happy to either keep it or delete it according to the editor’s recommendation.

Point 2: L243. Add “of the” before “topsoil”

Response 2: We have added “of the” before “topsoil” (L278 of the revised manuscript).

Point 3: L282. I cannot tell that the activation frequency of the sections between wells W3-W5 is always less than the upstream channels.

Response 3: The reviewer is right. We have rewritten that section as follows (L318-322 of the revised manuscript):

“Note that the activation frequency of the reach between wells W4 and W5 and wells W7 and W8 is less than that of the upstream channels in the Class 2 events. This also happens for wells W3 and W5 and W7 and W8 in the Class 3 events. These low frequency activation sites are the main sites where disconnection occurs. In addition to being the transition sections between ephemeral and intermittent flows, they are places prone to sediment (i.e. mainly sand) deposition in the streambed.”

Point 4: L298-299. I cannot understand this sentence.

Response 4: We have rewritten the sentence as follows (L341-342 of the revised manuscript):

“Those wells were dry or at their lowest level during the same events that the flow ceased in the intermittent channel reach.”

Point 5: L277. Have the authors thought of hyporheic flow? Was the groundwater in the context hyporheic flow? What is the role of hyporheic flow on the occurrence of overland flow? Have the authors thought of the influence of the channel morphology and riverbed microrelief on the connectivity? The slope between W5 and W7 (Figure 9a) is relatively steep compared to its downstream and upstream reach. Could the occurrence of runoff be attributed to the steep slope rather than shallow soil?

Response 5: We have thought about it but we were not able to measure hyporheic flow directly. Considering that the wells in the study area (Figure 9) are close enough to the channel (approximately 1.5m), we assume that the change in the water level in the well may be related to flow in the channel.

We believe that channel morphology and microrelief may play a role on the connectivity. However, we were not able to address this point at the moment as we have just started to address this issue in our field work.

We have improved the description of the average slope of the channels, as can be seen in L335-L337 of the revised manuscript. We agree that the slope influences the flow generation. On the other hand, we cannot affirm which characteristic has greater weight over the places where the flow remains longer. Between wells W11 and W10 the channel has the highest slope and the soil is shallow, even so the channel is ephemeral. Therefore, other characteristics such as the morphology and composition of the channel bed also have a strong influence. We added this discussion to the text in L331-335 of the revised manuscript.

Point 6: Figure 9a. Were the wells installed in the channel? If not, I was wondering if it is proper/meaningful to draw a line to connect all the data point.

Response 6: The reviewer is right, the wells in Figure 9 were not installed in the channel. They were installed at approximately 1.5m to the channel. We drew that line connecting them because we were trying to highlight the changes in soil depth which does not follow the topography exactly. We have improved Figure 9 and its description (L327-328 of the revised manuscript) to make it clear that wells are not connected and that we used the wells to extract soil depth information.

Point 7: Figure 9b. No data for W11?

Response 7: We could not measure water level in W11. Due to rock outcrops and very heterogeneous soil, we dug several holes around W11 and used the soil depth information only. We have tried to clarify this in L173-174 of the revised manuscript.

Round 2

Reviewer 1 Report

After reviewing the edits by the authors and their response to reviewers, I believe the changes to the manuscript have improved it substantially. I now believe the manuscript is acceptable for publication. There are minor grammatical errors, however, the scientific content and overall presentation of the results and discussion make it suitable for publication. 

Author Response

Response to Reviewer 1

After reviewing the edits by the authors and their response to reviewers, I believe the changes to the manuscript have improved it substantially. I now believe the manuscript is acceptable for publication. There are minor grammatical errors, however, the scientific content and overall presentation of the results and discussion make it suitable for publication.

Response: Thank you. We revised the text again for grammatical errors.